# Interactions between Mediterranean Diet Supplemented with Dairy Foods and the Gut Microbiota Influence Cardiovascular Health in an Australian Population

**DOI:** 10.3390/nu15163645

**Published:** 2023-08-19

**Authors:** Jocelyn M. Choo, Karen J. Murphy, Alexandra T. Wade, Yanan Wang, Ella L. Bracci, Courtney R. Davis, Kathryn A. Dyer, Richard J. Woodman, Jonathan M. Hodgson, Geraint B. Rogers

**Affiliations:** 1Microbiome Research and Host Health, Lifelong Theme, South Australian Health and Medical Research Institute, Adelaide, SA 5001, Australia; jocelyn.choo@sahmri.com (J.M.C.); geraint.rogers@sahmri.com (G.B.R.); 2College of Medicine and Public Health, Flinders University, Adelaide, SA 5042, Australia; 3Alliance for Research in Exercise, Nutrition and Activity, University of South Australia, GPO Box 2471, Adelaide, SA 5001, Australia; alexandra.wade@unisa.edu.au (A.T.W.); ella.bracci@mymail.unisa.edu.au (E.L.B.); courtney.davis@unisa.edu.au (C.R.D.); 4Clinical and Health Sciences, University of South Australia, GPO Box 2471, Adelaide, SA 5001, Australia; kate.dyer@unisa.edu.au; 5Allied Health and Human Performance, University of South Australia, GPO Box 2471, Adelaide, SA 5001, Australia; 6CSIRO MOSH-Future Science Platform, Health & Biosecurity, Adelaide, SA 5001, Australia; yanan.wang@csiro.au; 7Flinders Centre for Epidemiology and Biostatistics, Flinders University, GPO Box 2100, Adelaide, SA 5001, Australia; richard.woodman@flinders.edu.au; 8Nutrition and Health Innovation Research Institute, School of Medical and Health Sciences, Edith Cowan University, Perth, WA 6027, Australia; jonathan.hodgson@ecu.edu.au; 9Medical School, University of Western Australia, 35 Stirling Highway, Perth, WA 6000, Australia

**Keywords:** gut microbiome, Mediterranean diet, dairy foods, clinical trial

## Abstract

The impact of a Mediterranean diet on the intestinal microbiome has been linked to its health benefits. We aim to evaluate the effects of a Mediterranean diet supplemented with dairy foods on the gut microbiome in Australians at risk of cardiovascular disease. In a randomised controlled cross-over study, 34 adults with a systolic blood pressure ≥120 mmHg and with risk factors for cardiovascular disease were randomly allocated to a Mediterranean diet with 3–4 daily serves of dairy foods (Australian recommended daily intake (RDI) of 1000–1300 mg per day (MedDairy)) or a low-fat (LFD) control diet. Between each 8-week diet, participants underwent an 8-week washout period. Microbiota characteristics of stool samples collected at the start and end of each diet period were determined by 16S rRNA amplicon sequencing. MedDairy-associated effects on bacterial relative abundance were correlated with clinical, anthropometric, and cognitive outcomes. No change in the overall faecal microbial structure or composition was observed with either diet (*p* > 0.05). The MedDairy diet was associated with changes in the relative abundance of several bacterial taxa, including an increase in *Butyricicoccus* and a decrease in *Colinsella* and *Veillonella* (*p* < 0.05). Increases in *Butyricicoccus* relative abundance over 8 weeks were inversely correlated with lower systolic blood pressure (r = −0.38, *p* = 0.026) and positively correlated with changes in fasting glucose levels (r = 0.39, *p* = 0.019), specifically for the MedDairy group. No significant associations were observed between the altered taxa and anthropometric or cognitive measures (*p* > 0.05). Compared to a low-fat control diet, the MedDairy diet resulted in changes in the abundance of specific gut bacteria, which were associated with clinical outcomes in adults at risk of CVD.

## 1. Background

Diet is a fundamental determinant of metabolic health and immune regulation. There is clear evidence linking the impact of dietary patterns on the development of metabolic diseases, including obesity [1], type 2 diabetes [2], and cardiovascular diseases [3]. These conditions are often associated with a cluster of interrelated metabolic phenotypes, including insulin resistance, abnormal glucose regulation, lipid dysregulation (high triglycerides and low high-density lipoprotein cholesterol), hypertension, and chronic systemic inflammation [4,5,6]. 

Long-term dietary patterns also play an important role in shaping the intestinal commensal microbiota [7]. In turn, the microbial production of bioactive metabolites through the metabolism of dietary components can influence host physiology, including metabolic and immune homeostasis [7,8]. For example, the production of short-chain fatty acids (SCFAs), including acetate, propionate, and butyrate, through the fermentation of non-digestible fibre, provides a source of energy for colonic tissues, promotes the maintenance of gastrointestinal tissue integrity, modulates inflammatory pathways, and suppresses the growth of pathogenic bacteria [8]. In contrast, the microbial metabolism of choline and L-carnitine, which are high in eggs, fish, meat, nuts, and dairy, results in the production of trimethylamine N-oxide (TMAO), a metabolite that has been associated with inflammation and atherosclerosis [7,9]. 

The Mediterranean diet (MedDiet) is characterised by high consumption of vegetables, fruits, nuts, legumes, cereals, and extra virgin olive oil (EVOO); moderate consumption of fish, eggs, poultry, dairy foods, and red wine; and low consumption of red meat (animal proteins) and discretionary foods, including cakes, sweets, and fast foods [10]. The MedDiet delivers a range of bioactive nutrients, including antioxidants, fibre, vitamins and minerals, polyphenols, monounsaturated fats, and omega-3 polyunsaturated fats [11], many of which can promote beneficial health effects via the gut microbiota [12,13]. Changes in the levels of specific gut bacteria and microbiome function are associated with improvements in cardiometabolic biomarkers and immune regulation, including lower levels of triglycerides [12], improved liver function [13], and a reduction in systemic inflammatory markers, such as C-reactive protein [14]. 

Epidemiology-based studies have reported that higher adherence to the MedDiet is associated with lower prevalence of central obesity, dyslipidemia, metabolic syndrome, and a reduced risk of diabetes [15,16]. Additionally, diet intervention studies, including an 8-week MedDiet in overweight or obese adults [12], a 3-month low-calorie MedDiet intervention in adults at risk of cardiovascular disease (CVD) (CARDIVEG study) [13], and a 12-month MedDiet for elderly populations (NU-AGE study) [14], have all demonstrated that a Mediterranean-based diet modulates the gut microbiome towards a state that promotes beneficial effects on metabolic health and reduces the risk of CVD. 

A traditional MedDiet provides approximately 700–820 mg of calcium per day, which is below the Australian recommended daily intake (RDI) of 1000–1300 mg per day [17,18,19]. Calcium is an essential mineral not only for the formation and maintenance of bones, but also for other biological processes, including cell signalling for muscle function, vascular dilation and contraction [20], neuronal activity [21], and cell differentiation [22]. Inadequate calcium intake is related to reduced bone strength, pregnancy complications such as hypertension and pre-eclampsia [23], and an increased risk of cardiovascular disease [24].

The recommended calcium intake varies between populations, in accordance with health-related behaviours and the relative risk of calcium-related health outcomes [25]. Before a MedDiet is recommended as a long-term dietary pattern, it should meet the nutritional requirements of the target population. Supplementing the MedDiet with dairy products, such as milk, cheese, and yoghurt, that are rich in calcium would ensure the MedDiet aligns with Australian calcium intake recommendations. These foods are also capable of altering the gut microbiome [26,27], promoting the expansion of *Bifidobacterium* species and lactic acid bacteria with a reciprocal relative depletion of other commensal bacteria, including *Faecalibacterium prausnitzii*. Bifidobacteria and lactic-acid bacteria are known to possess B-group vitamin biosynthesis capacity, which is required for cellular enzymatic and metabolic processes [28], while *Faecalibacterium prausnitzii* can confer protective function due to its anti-inflammatory effects and role in maintaining intestinal permeability [26].

We undertook a MedDairy dietary intervention trial in an Australian adult population at risk of CVD to evaluate whether a MedDiet enriched with dairy products promotes beneficial cardiometabolic and cognitive health outcomes [29]. Specifically, a traditional Mediterranean diet was modified to meet the Australian RDI of calcium (between 1000 and 1300 mg) through increased intake of dairy foods [29]. The MedDairy trial [29,30] involved a randomised controlled cross-over design study, where participants adhered to an 8-week MedDairy and an 8-week control non-Mediterranean low-fat diet (shown to improve CVD risk [12] and is termed hereafter as a low-fat diet). The MedDairy intervention was found to be associated with significant improvements in cardiovascular risk, including lower clinic and morning systolic blood pressure (monitored during clinic visits and self-monitored at home, respectively), lower morning diastolic blood pressure, lower triglyceride levels, an increase in HDL cholesterol, and a lower total cholesterol:HDL ratio [29]. In the current study, we aim to investigate the impact of the Mediterranean diet enriched with dairy products on the intestinal microbiome and to evaluate the relationships between MedDairy diet-specific changes in the gut microbiota and the clinical outcomes of the original trial [29].

## 2. Materials and Methods

### 2.1. Study Overview

A randomised controlled trial with a 2 × 2 cross-over design was used to compare a Mediterranean diet supplemented with adequate dairy foods (MedDairy or MD) against a low-fat (LFD) control diet in individuals at risk of cardiovascular disease (CVD) (study protocol has been described previously [30]). This trial was registered in the Australia and New Zealand Clinical Trials Register (ACTRN12616000309482) and approved by the University of South Australia Human Research Ethics Committee (#34954 on 17 December 2015).

### 2.2. Participants 

The MedDairy study recruited adults with a risk of cardiovascular disease (CVD). Inclusion criteria were age between 45 and 75 years, an elevated systolic blood pressure (SBP) of ≥120 mmHg, not on hypertensive medication, and a diagnosis of at least two of the following: overweight/obesity (body mass index, BMI ≥ 25 kg/m^2^); waist circumference > 94 cm for men and >80 cm for women; total cholesterol ≥ 5.5 mM; triglycerides (≥2.0 mM); low-density lipoprotein (LDL) (≥3.5); or high-density lipoprotein (HDL) ≤ 0.9 for men and ≤1.0 for women; impaired glucose tolerance (6.1–7.8 mmol/L); and/or a family history of CVD or type 2 diabetes. Individuals that consume medicinal levels of calcium or >1000 mg of omega-3 supplements daily were excluded from the study. Additional inclusion and exclusion criteria are reported elsewhere [30].

### 2.3. Study Intervention

Participants followed a MedDairy or LFD diet intervention for 8 weeks, separated by an 8-week washout phase where participants returned to their habitual diet. Participants were stratified by gender and age and randomly assigned to their first dietary phase (MedDairy or LFD) by a blinded independent staff member. Instructions on how to adopt each diet were provided by a dietitian at the start of each dietary phase. Detailed descriptions of the MedDairy and LFD dietary phases and instructions are as previously reported [30]. Briefly, components of the Mediterranean diet were primarily focused on consumption of fresh fruits, vegetables, legumes, fish, seafood, nuts, seeds, whole grain cereal products, selected white meats (poultry without skin), limited or non-consumption of red meat, processed meats, cream, butter, sugared beverages or bakery items, and the use of extra virgin olive oil (EVOO) for cooking or salad dressing. The recommended dairy consumption was 3–4 daily servings of dairy food based on the following: one serve = 250 mL low-fat milk, 40–120 g hard and/or semisoft to soft cheese, 200 g low-fat Greek yoghurt, or 200 g tzatziki dip. This serving size of dairy foods delivers approximately 900–1200 mg of calcium, while vegetables and nuts provide approximately 100 mg of calcium. The food products consumed are based on an Australian food supply, and participants were not restricted to dietary sodium intake as it is not a component of the Mediterranean diet. 

The low-fat diet was modelled after the PREDIMED trial [29], in which participants were instructed to follow their habitual diet but reduce their total fat intake. The use of a low-fat diet has previously been recommended for heart health, particularly around the time of the design of PREDIMED. Participants were instructed to consume low-fat food (breads, cereals, lean meat, legumes, rice, vegetables, and fruits) and choose low-fat variations of food products (such as low-fat dairy) as a replacement to restrict or avoid high-fat foods, including cream, full fat dairy, processed meats, high fat meats, nuts, ice cream, and sugared bakery items. Participants are also advised to consume no more than 20 mL of any oil type, no more than two teaspoons of butter or margarine daily, and to remove visible fat and skin from meat and fish before cooking. Participants were not recommended a set dairy consumption as the low-fat diet was intended to match their habitual dietary intake. 

### 2.4. Study Assessment and Outcomes

Eligible participants completed a clinic assessment visit and faecal sample collection (for gut microbiome analysis) at the start (weeks 0 and 16) and at the end (weeks 8 and 24) of each diet intervention phase. To determine the effect of a MedDairy diet on the gut microbiota, changes to the gut microbiota between baseline of the intervention phase and at the end of the 8-week diet intervention, for the MedDairy or LFD diet, were assessed. Between-group differences at the end of the 8-week diet intervention were also assessed to determine MedDairy diet-specific effects on bacterial relative abundance. 

The exploratory outcomes were associations between changes in MedDairy diet-specific bacterial relative abundance and changes in clinical, anthropometric, and cognitive function measurements. Clinical measures include blood pressure (systolic and diastolic blood pressure at a clinic or self-measured), blood levels of insulin, glucose, HOMA-IR, C-reactive protein, total triglyceride, total cholesterol, HDL, LDL, and cholesterol:HDL ratio [29,30]. Anthropometric measures include BMI, weight, and body composition assessed via dual-energy X-ray absorptiometry (lean mass, fat mass, and abdominal fat mass) [27,28]. Cognitive function includes planning, memory, attention, processing speed, and Addenbrook’s Cognitive Examination (ACE-R) scores [29,30,31]. Correlations between variables were performed based on the changes during both the MedDairy and low-fat diet periods. Significant associations were then further assessed for diet-specific significance by stratifying according to the MedDairy and low-fat diet periods only.

### 2.5. Dietary Adherence

Dietary adherence was assessed and reported previously [30,32]. Briefly, a 14-item MedDiet adherence tool and a 9-item low-fat diet adherence tool were completed for each fortnight during the appropriate intervention. A 3-day weighed food record (WFR) was used to calculate the Mediterranean diet score (MDS) as an indicator of MedDiet adherence, using a 10-point [19] and 18-point [33] scoring system. Participants adhered to both dietary interventions, with increased adherence to the MedDiet during the MedDairy intervention, as assessed using the 10-point and 18-point MDS (Appendix A) [29].

### 2.6. Faecal Microbiome Analysis

Faecal sample collection was performed in an OMNIgene GUT stool collection kit (DNA Genotek, Ottawa, ON, Canada) for sample preservation, according to instructions provided by the manufacturer, and using a disposable urine collection hat (EBOS Healthcare, Kingsgrove, NSW, Australia). Samples were returned to the laboratory within approximately 24 h of collection and stored at −80 °C until analyses. Faecal DNA extraction was performed by transferring 200 µL of the faecal slurry to a tube containing 200 µL of 1X phosphate buffered saline (PBS, pH 8.0) (Life Technologies, Waltham, MA, USA). The faecal slurry was mixed by vortexing and centrifuged at 13,000× *g* for 20 min. DNA was extracted from faecal pellets using the DNeasy PowerLyzer PowerSoil Kit (QIAGEN, Straße, Hilden, Germany), according to the manufacturer’s instructions with modifications as previously described [34]. Total bacterial load was assessed by quantitative PCR as described previously [34] to establish that the levels were similar at baseline and between paired samples following either diet intervention. Amplicon libraries of the 16S rRNA V4 hypervariable region were generated and indexed for sequencing using an Illumina Miseq v3 kit on an Illumina Miseq platform at the South Australian Genomics Centre (Adelaide, SA, Australia) as previously described [35].

Paired-end sequence reads were demultiplexed, and bioinformatics analysis was performed using the QIIME2 platform (release 2021) for microbiota profiling [36], using a bioinformatics pipeline as previously described [37]. Taxonomic assignment was performed in reference to the SILVA 16S ribosomal database (version 138) which clustered at 99% sequence identity [38]. All samples were subsampled to the lowest sample read depth of 7794 sequence reads for microbiome analysis.

Microbial alpha diversity measures of richness, evenness, and diversity (observed species, Pielou’s evenness, and Faith’s phylogenetic diversity, respectively), as well as weighted Unifrac distances for composition analysis, were computed using QIIME2. Raw sequence data are publicly accessible from the Sequence Read Archive repository (BioProject ID: PRJNA996323). 

### 2.7. Statistical Analyses

Baseline participant demographics between dietary groups were statistically compared using an unpaired *t*-test for continuous variables. Data were analysed using STATA (Version 13, StataCorp, College Station, TX, USA) and IBM SPSS Statistics (Version 21). Residuals were screened for normality, and non-normal variables were transformed through the use of log10 and square root transformations.

All microbiome data were checked for normality using the Shapiro–Wilks method. Changes in overall microbial alpha diversity measures were analysed using the Wilcoxon sign-rank or Mann–Whitney *t*-test for paired and unpaired data, respectively. Microbiota composition differences based on the weighted Unifrac distances were analysed using a permutation-based ANOVA package (PERMANOVA) on PRIMER (v7). PERMANOVA analyses were performed with diet and time as the main factors, with subjects nested in these main factors. A statistical comparison of the bacterial taxa’s relative abundance between the diet groups was performed using a Wilcoxon sign-rank or Mann–Whitney *t*-test for paired and unpaired data, respectively. Type I errors in multiple comparisons were adjusted using the false discovery rate method. All statistical comparisons were assessed based on a significance level of *p* < 0.05. 

The association between changes in specific bacterial relative abundance and changes in clinical, anthropometric, and cognitive assessment outcomes was performed using a repeated measures correlation analysis package in R (rmcorr v.0.5.4). Similar correlations were performed to determine associations between bacterial relative abundance and adherence to the MedDiet based on the 10- and 18-point MDS.

### 2.8. Power Calculation

The initial sample size calculation was based on the ability to detect a clinically relevant difference of 2.5 mmHg in systolic blood pressure (primary study outcome) at a power of 90% and *p* < 0.05, as previously reported [30]. Briefly, a sample size of 31 volunteers was required, assuming a within-group SD of 14 mmHg, a within-subject correlation between the 4 BP measures at each visit of r = 0.6, and a between-phase within-subject correlation of ρ = 0.5. This correlation (ρ) and the crossover design reduced the number of required participants by a factor of 1/[(1 − ρ)/2], that is, from *n* = ∼124 for a parallel-group design with the use of ANCOVA (*n* = 62/group) to *n* = 31 subjects in total [39]. For the current study, the final sample size of *n* = 34 subjects provided *n* = 68 measures of change across the 2 periods. This provided 80% power to detect Pearson, Spearman, and Kendall’s tau correlation coefficients of r = 0.32, 0.34, and 0.34, respectively at a 2-sided type 1 error rate of alpha = 0.05. Based on the observed mean differences between diets and the standard deviation of these differences, we also had 80% power to detect differences in richness (mean Δ = 1.68, SD Δ = 2.69), Pielou (mean Δ = 0.0031, SD Δ = 0.0020), and Faith’s phylogenetic diversity (mean Δ = 0.78, SD Δ = 0.25).

## 3. Results

Of the 43 participants randomised, 41 commenced the diet schedule and 37 continued to completion (Figure 1). Participant withdrawal was not associated with either the MedDairy or LFD interventions, with two participants dropping out during each arm. Complete faecal and clinical data were available to perform gut microbiome analyses on paired samples at baseline and 8 weeks for 34 participants. At baseline, there were no significant differences between Group 1 (MedDairy intervention first, *n* = 18) and Group 2 (LFD intervention first, *n* = 16) for age, the distribution of gender, or the level of education (Table 1). 

### 3.1. Effects of Diet Intervention on the Overall Faecal Microbiota Structure and Composition

No significant changes in bacterial richness and evenness were evident after the MedDairy or LFD diet study arms (Figure 2A,C,E), and these measures did not differ significantly between the MedDairy and LFD groups at the end of the 8-week diet intervention (Figure 2B, 2D and 2F, respectively). Microbial diversity was not altered by the MedDairy diet but was significantly lower following the LFD intervention (*p* = 0.037) (Figure 2E). This change in the LFD group was modest, as microbial diversity did not significantly differ between the MedDairy and LFD groups at the end of the 8-week diet intervention (Figure 2F). An assessment of the gut microbiota composition shows no significant changes following the MedDairy or LFD (PERMANOVA *p* = 0.573 and *p* = 0.249, respectively) (Appendix A). Additionally, the microbiota composition of the MedDairy group did not differ significantly from the LFD group at the end of the 8-week diet intervention (*p* = 0.99). Together, the results suggest that the MedDairy diet, or LFD, does not have broad effects on the gut microbial community. 

The potential for effects resulting from the MedDairy or LFD intervention at Phase 1 to affect baseline measures at Phase 2 (following the 8-week washout period) was investigated. Carryover effects from Phase 1 were assessed by comparing baseline measures of paired samples at Phase 1 to those at Phase 2 (corresponding to Week 0 and Week 15 of the study, respectively). Bacterial richness, evenness, and diversity did not differ significantly between the Phase 1 baseline and the Phase 2 baseline (Appendix A). A similar result was observed when this comparison was based on the MedDairy or LFD intervention groups (Appendix A). Additionally, microbiota composition did not significantly differ between the baseline of Phase 1 and Phase 2 periods (PERMANOVA *p* = 0.943), or when stratified to the MedDairy and LFD intervention groups (*p* = 0.082 and *p* = 0.248, respectively). Together, these results indicate the absence of significant carryover effects between phases. The overall (Phase 1 and Phase 2 combined) baseline alpha diversity (Appendix A) and microbiota composition (PERMANOVA *p* = 0.985) also did not differ significantly between the MedDairy and LFD intervention groups.

### 3.2. Differential Effects between MedDairy and Low-Fat Diets on Specific Gut Bacteria

While the broad characteristics of the faecal microbiota remained unaltered by MedDairy or LFD, changes were observed at the level of individual bacteria taxa. Following the 8-week MedDairy intervention, significant changes were observed for several bacterial taxa, including increases in the relative abundance of *Butyricicoccus*, Lachnospiraceae NK4A136, and *Streptococcus*, as well as decreases in the relative abundance of *Colinsella*, *Veillonella*, two taxa in the Oscillospiraceae family (Oscillospirales uncultured, UCG-002), and Ruminococcaceae UBA 1B19 (Figure 3, Appendix A). Additionally, MedDairy and LFD were associated with significant but opposite changes in *Collinsella*, which decreased in relative abundance with the MedDairy diet but increased in relative abundance with the LFD diet (Figure 3). Between diet group comparisons indicate that the relative abundance of *Butyricicoccus*, *Veillonella*, and *Collinsella* significantly differed between the MedDairy and LFD groups at the end of the 8-week diet intervention, but these differences were not observed at baseline (Figure 3, Appendix A), suggesting that the gut microbiota changes were mediated primarily by the MedDairy diet. In contrast, the relative abundance of *Lachnospiraceae NK4A136*, *Streptococcus*, *Oscillospiralles* (uncultured and UCG-002), and *Ruminococcaceae UBA1819* did not significantly differ between the MedDairy and LFD groups at the end of the 8-week diet intervention, suggesting that the changes in these taxa were modest or less specific to the MedDairy diet (Figure 3, Appendix A). Notably, the bacterial taxa changes described were significant only when uncorrected for multiple comparisons. 

### 3.3. Gut Bacteria Changes Were Associated with Diet and Clinical Outcomes 

Associations between changes in altered bacterial taxa and changes in clinical, anthropometric, and cognitive measures were determined by correlation analysis (Figure 4). The relationship between changes in bacterial taxa and adherence to the MedDairy diet was also assessed. Bacterial taxa selected for correlation analysis were taxa that differed following 8 weeks of the MedDairy diet and also differed between the diet intervention groups, including *Collinsella*, *Butyricicoccus*, and *Veillonella*.

Correlation analysis based on both interventions (MedDairy and LFD) indicates that a higher relative abundance of *Butyricicoccus* was significantly associated with higher adherence to a MedDiet (10-point score: r = 0.28, *p* = 0.003, and 18-point score: r = 0.33, *p* < 0.001) (Figure 4D,E, respectively). These associations remained significant following FDR correction for multiple comparisons (FDR *p* < 0.05) (Appendix A). A stronger and more significant association between *Butyricicoccus* and adherence to the MedDairy diet was observed within the MedDairy period only (10-point score: r = 0.46, *p* = 0.006; 18-point score: r = 0.51, *p* = 0.002), while no significant associations were observed for the LFD period (10-point score: r = 0.20, *p* = 0.246; 18-point score: r = 0.20, *p* = 0.252). 

The changes in relative abundance of *Butyricicoccus* were significantly correlated with the changes in systolic blood pressure (SBP clinic) and fasting glucose levels for the combined MedDairy and LFD intervention periods only when assessed without FDR correction for multiple testing (Figure 4A, Appendix A). In particular, increases in the relative abundance of *Butyricicoccus* were associated with a decline in systolic blood pressure (r = −0.25, *p* = 0.013), but were positively associated with fasting glucose levels (r = 0.23, *p* = 0.019) (Figure 4B,C, respectively). When stratified by diet, these associations were observed for the MedDairy group (SBP: r = −0.38, *p* = 0.026; Glucose: r = 0.39, *p* = 0.019) and not for the LFD group (SBP: r = −0.20, *p* = 0.235; Glucose: r = 0.07, *p* = 0.683). While significant correlations were observed between *Collinsella* and the Cholesterol:HDL ratio (r = 0.22, *p* = 0.027), as well as to the 18-point MDS (r = −0.30, *p* = 0.002) (Figure 4A) when considering the whole cohort, these effects were not observed for the MedDairy group (Cholesterol:HDL ratio: r = −0.17, *p* = 0.350; 18-point score: r = −0.33, *p* = 0.058). None of the taxa changes were associated with cognitive measures including planning, memory, attention, processing speed, and ACE-R scores (*p* > 0.05).

## 4. Discussion

This study sought to explore the effects of an 8-week Mediterranean diet enriched with dairy products (to reach the recommended Australian dietary intake of 1000–1300 mg of calcium) on the gut microbiome of Australian adults at risk of CVD. The dairy enrichment involved 3–4 daily servings of any of the following dairy foods: one serve (250 mL) of low-fat milk, 40–120 g of hard and/or semisoft to soft cheese, 200 g of low-fat Greek yoghurt, or 200 g of tzatziki dip. Participants recruited to the study recorded a MedDiet adherence score at moderate levels based on their habitual diet at baseline (average ± SD of 4.4 ± 1.8 for the 10-point score and 7.0 ± 1.4 for the 18-point score), which reflect those of an Australian population that are not following a prescribed MedDiet [40,41,42], and an increased MedDiet adherence during the MedDairy intervention [29].

Faecal microbiota analysis was performed based on paired comparisons between baseline and after 8 weeks of MedDairy or LFD intervention and between the groups at 8 weeks. Paired analysis indicated that the overall structure and composition of the faecal microbiota were not significantly altered by the MedDairy or LFD groups, except for a modest decrease in microbial diversity in the LFD group. These results are consistent with other studies that assessed the gut microbiome following a MedDiet intervention, including the CARDIVEG study, which involved a 3-month low-calorie MedDiet in adults with CVD risk [13], and the NU-AGE study, a 12-month MedDiet tailored for elderly subjects [14]. Each of these studies reported that the broad gut microbiota composition was unaltered by the MedDiet intervention, although significant changes in the relative abundance of several bacterial taxa were observed. Similar observations were reported in cross-sectional studies of healthy adult populations that were classified according to MedDiet scores for habitual dietary patterns. In particular, the Men’s Lifestyle Validation Study, comprising a cohort of 307 healthy men [43], and another study in healthy young adults of low (MDS: 0–5) and high (6–9) MDS groups [44], indicated a modest effect of MedDiet on the overall structure of the gut microbiome and microbial diversity specifically. 

The MedDairy diet did not result in broad changes to the gut microbiota but significantly altered the relative abundance of selected bacterial taxa, a result that is consistent with previous MedDiet studies [13,14]. Specifically, the MedDairy diet resulted in significant increases in the relative abundance of *Butyricicoccus* and a decrease in *Colinsella* and *Veillonella*. Previous short-term diet intervention studies indicated that *Butyricicoccus* levels increased significantly within 2–4 days of the start of a MedDiet intervention [13,45], but decreased upon switching to a non-Mediterranean diet [45,46]. However, the levels of *Butyricicoccus* were not reported to be altered in longer-period diet intervention studies of 3-months [13] or 1-year MedDiet [14]. Given the rapid response of *Butyricicoccus* to dietary components, our results suggest that the increase in *Butyricicoccus* relative abundance can be sustained by 8 weeks of a MedDiet specifically, as observed with the MedDairy group, although it is possible that such effects may diminish with long-term MedDiet consumption. 

Reductions in the levels of *Collinsella* [12,14,45] and *Veillonella* following the MedDairy diet were also consistent with previous MedDiet studies [12,14,47]. An increase in levels of *Collinsella* is linked to low dietary fibre intake [48], with the bacteria found to be associated with detrimental effects on metabolic regulation [48] and intestinal permeability [49]. A higher abundance of *Collinsella* has been reported in metabolic diseases, including CVD [50], type 2 diabetes [51], and rheumatoid arthritis [49], suggesting that its reduction following the MedDairy diet intervention may contribute to beneficial health outcomes. *Veillonella* is known to produce SCFA acetate and propionate from the fermentation of lactate [52], although its causal role in diseases has not been determined.

The MedDairy diet intervention was associated with modest but significant increases in Lachnospiraceae and *Streptococcus* and reductions in Oscillospiraceae and Ruminococcaceae. Significant changes in the abundance of Lachnospiraceae, Oscillospiraceae, and Ruminococcaceae families have been previously reported in MedDiet studies [12,14,44], although these studies also reported alterations in bacterial genera, including increases in *Faecalibacterium praunitzii*, *Roseburia*, and *Eubacterium* [12,14], as well as decreases in *Parabacteroides* [12,13], *Coprococcus*, and *Dorea* [14], which were not observed in our study. These results suggest that selected components of the MedDairy diet may mediate a more pronounced effect on specific bacterial taxa. For example, *Streptococcus thermophilus*, a bacterial marker of dairy product consumption, was previously found to decrease with the MedDiet [12]. The increase in *Streptococcus* in the MedDairy group may be attributed to the higher dairy intake in this study. While *Streptococcus thermophilus* has been linked to anti-inflammatory effects [53,54], other *Streptococcus* species are associated with low-grade inflammation [55] and colorectal cancer (*Streptococcus bovis*) [56].

Higher adherence to a MedDiet diet has been previously shown to mediate protective effects on cardiometabolic risk factors, including blood pressure (lowers systolic and diastolic blood pressure), lipids (lower total cholesterol:HDL ratio, lower LDL levels, and higher HDL levels), and/or glycaemic control (lowers insulin secretion and HOMA-IR levels) when compared to the group consuming a healthy dietary guideline diet [12,57]. We observed a similar effect with the MedDairy diet [29]. A higher abundance of *Butyricicoccus* was significantly associated with lower SBP but increased fasting blood glucose. This association was observed only for the MedDairy group, but not the LFD group, suggesting an effect specifically directed by the MedDairy diet. Recent studies suggest that blood pressure can be influenced by an increased inflammatory state and intestinal permeability [58]. Beneficial effects on immune regulation and intestinal permeability can be mediated by bacterial metabolites, such as SCFAs, while other bacterial-derived metabolites and components, such as trimethylamine (TMA) and lipopolysaccharide, respectively, are associated with inflammatory properties [59]. Reduced levels of *Butyricicoccus*, a SCFA producer, have been associated with chronic gastrointestinal and kidney diseases [60,61]. *Butyricicoccus* has also been shown to provide therapeutic potential for gastrointestinal inflammation, including ulcerative colitis and Crohn’s disease, through its protective effects on the integrity of intestinal epithelial cells to maintain intestinal permeability [61,62]. These findings suggest the potential for *Butyricicoccus* to beneficially modulate blood pressure, as observed in the MedDairy group. A higher abundance of *Butyricicoccus* is also positively correlated with higher fasting plasma glucose levels but was not significantly associated with other determinants of glucose homeostasis, including insulin and HOMA-IR levels. Given that glucose regulation can also affect blood pressure [63], more frequent and detailed assessment of changes in the gut microbiome, blood pressure, and blood glucose measurements during the diet intervention period will be necessary to establish the role of gut microbiota, including *Butyricicoccus*, to mediate such interactions. Previous studies have also shown that levels of *Butyricicoccus* increased during ageing and replaced the loss of several commensal bacteria [64]. It is therefore possible that the impact of a MedDairy diet on *Butyricicoccus*, which was observed in this study, may differ in other cohorts with different age demographics. 

Our study was limited to the measurement of gut microbiota levels and clinical outcomes but did not measure other factors such as changes in metabolic capacity [12] or the level of physical activity [43], which can influence cardiometabolic health. The gut microbiota changes we report were based on those mediated by a Mediterranean-based diet with dairy supplementation rather than a specific dietary component. Further, the use of a low-fat diet control group was appropriate at the time of the study design; however, it may no longer be the most suitable control choice given that dietary guidelines support the change in diet quality and the quality of dietary fats as a strategy for heart health. 

Taken together, our results suggest that an 8-week MedDiet supplemented with dairy foods results in relative abundance changes in bacterial taxa. Microbial changes, including an increase in *Butyricicoccus*, were inversely correlated with changes in systolic blood pressure, a clinical measure that was previously shown to be reduced following the MedDairy diet.

## Figures and Tables

**Figure 1 nutrients-15-03645-f001:**
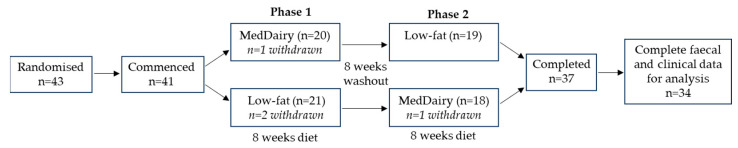
Flow diagram of the cross-over study involving an 8-week MedDairy and an 8-week low-fat diet intervention in adults with cardiovascular disease risk. Adapted from Wade et al., 2018 [27].

**Figure 2 nutrients-15-03645-f002:**
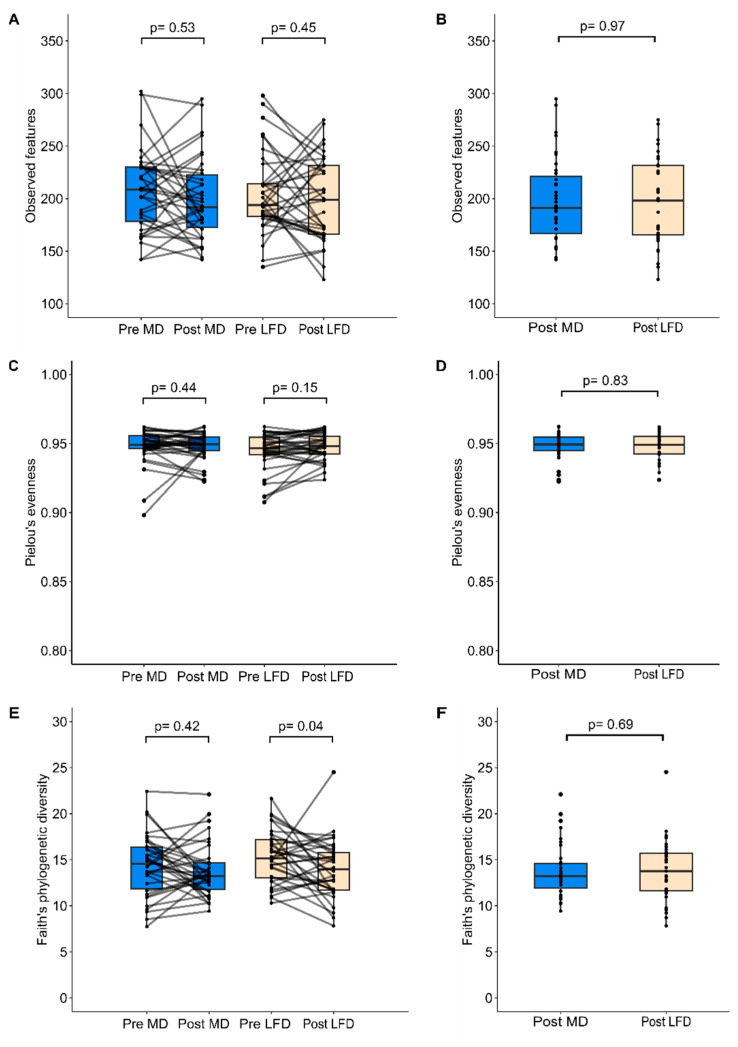
Alpha diversity measures of (**A**,**B**) microbial richness, (**C**,**D**) evenness, and (**E**,**F**) diversity were assessed before and after 8 weeks of MedDairy (MD) or low-fat (LFD) diet intervention. Statistical comparisons were performed using the Wilcoxon sign-rank test (within-group assessment) or Mann–Whitney test (between-group assessment) at a significance level of *p* < 0.05.

**Figure 3 nutrients-15-03645-f003:**
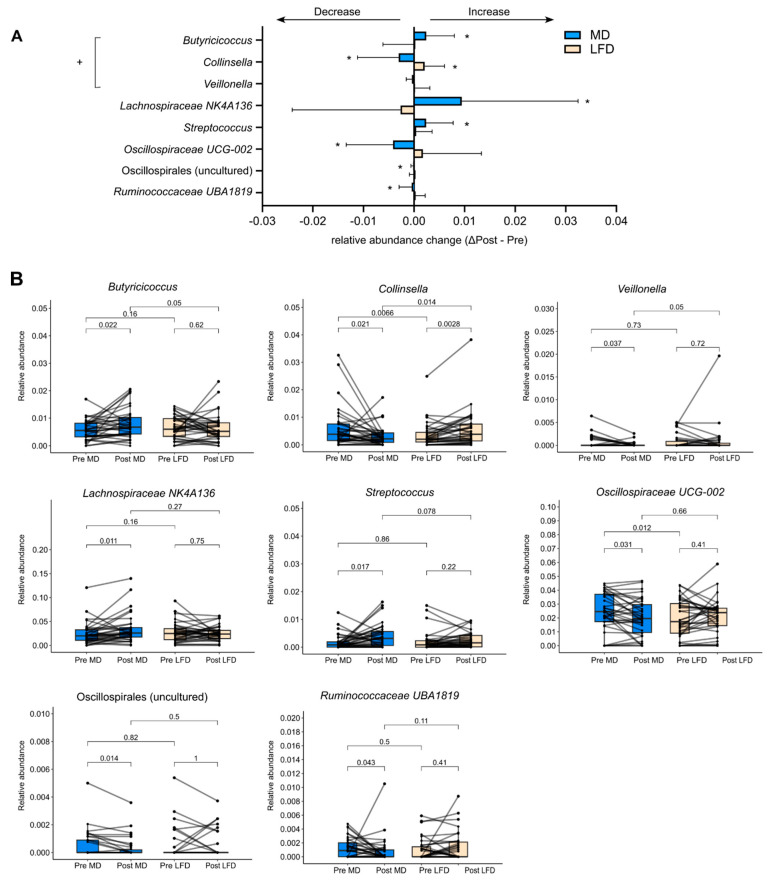
Alterations in bacterial relative abundance following 8 weeks of the MedDairy diet. (**A**) Bacterial taxa that were significantly altered by 8 weeks of the MedDairy (MD) diet (Post MD–Pre MD), as well as the corresponding taxa relative abundance changes for the low-fat (LFD) diet group, are shown. Statistical comparisons were performed using the Wilcoxon sign-rank test (within-group assessment) or the Mann–Whitney test (between-group assessment). A significance level of *p* < 0.05 is indicated by an asterisk (*). Bacterial relative abundance changes that are significant for within- and between-group comparisons are indicated as +. (**B**) Paired analysis of significant taxa in the MedDairy and low-fat diet groups.

**Figure 4 nutrients-15-03645-f004:**
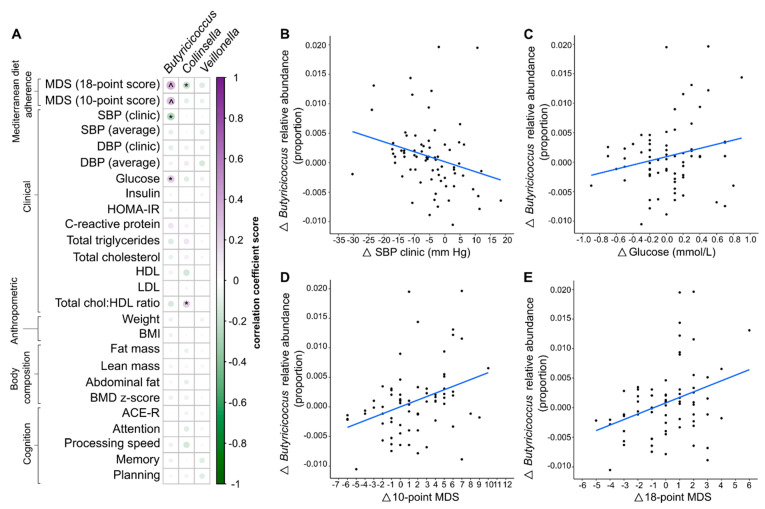
(**A**) Spearman’s correlation between significantly altered taxa and Mediterranean diet adherence scores, clinical, anthropometric, body composition, and cognitive measurements. ^ denotes statistical significance based on a false discovery rate-adjusted *p*-value of <0.05, and * denotes *p* < 0.05. (**B**–**E**). Only correlations that were significant for both the overall participant cohort and the MedDairy group were plotted. Relative abundance changes of *Butyricicoccus* (post-pre diet intervention) were plotted against the corresponding changes in clinical and Mediterranean diet adherence scores. The correlation dataset shown represents the overall participant cohort.

**Table 1 nutrients-15-03645-t001:** Demographic and clinical characteristics of the study sample at week 0 (start of the study trial) by group.

	Group 1 (*n* = 18)	Group 2 (*n* = 16)	Total (*n* = 34)
Age (years)	62.0 (6.2)	60.2 (7.7)	61.1 (6.9)
Gender			
Males	5	5	10
Females	13	11	24
Education (years)	15.3 (2.9)	17.3 (4.7)	16.2 (3.9)
Home SBP average (mmHg)	127.3 (14.8)	128.8 (14.5)	128.1 (14.6)
Home DBP average (mmHg)	77.9 (9.4)	78.1 (10.5)	78.0 (9.9)
Home HR average (mmHg)	71.9 (10.1)	71.3 (10.1)	71.6 (10.0)
Clinic SBP (mmHg)	130.8 (13.1)	132.8 (11.6)	131.7 (12.2)
Clinic DBP (mmHg)	84.6 (11.0)	91.1 (10.5)	87.7 (11.1)
Clinic HR (bpm)	68.5 (11.1)	65.9 (7.9)	67.3 (9.7)
Insulin (mU/L)	14.4 (7.9)	10.4 (4.5)	12.7 (6.8)
Glucose (mmol/L)	5.8 (0.5) *	5.4 (0.5)	5.6 (0.5)
Total triglycerides (mmol/L)	1.7 (1.3)	1.4 (0.6)	1.6 (1.0)
Total cholesterol (mmol/L)	5.9 (1.1)	5.5 (1.0)	5.7 (1.1)
HDL (mmol/L)	1.5 (0.4)	1.5 (0.4)	1.5 (0.4)
LDL (mmol/L)	3.7 (1.0)	3.4 (0.8)	3.5 (0.9)
Cholesterol/HDL ratio	4.4 (1.7)	3.8 (1.0)	4.1 (1.4)
Weight (kg)	86.1 (15.8)	85.3 (13.4)	85.8 (14.5)
Height (m)	1.7 (0.1)	1.7 (0.1)	1.7 (0.1)
BMI (kg/m^2^)	30.5 (3.9)	30.6 (3.0)	30.5 (3.5)

Values are presented as mean (SD). * Significant difference between groups at *p* < 0.05. Group 1 = MedDairy first; Group 2 = Low-Fat first. SBP, systolic blood pressure; DBP, diastolic blood pressure; HR, heart rate; mmHg, millimetres of mercury; bpm, beats per minute. SBP, systolic blood pressure; DBP, diastolic blood pressure; HR, heart rate; CRP, C-reactive protein; HDL, high-density lipoprotein; LDL, low-density lipoprotein; BMI, body mass index.

## Data Availability

Raw sequence data are publicly accessible from the Sequence Read Archive repository using the BioProject ID PRJNA996323 and at https://www.ncbi.nlm.nih.gov/bioproject/996323, accessed on 18 August 2023.

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
