# Peer review of "Interactions between Mediterranean Diet Supplemented with Dairy Foods and the Gut Microbiota Influence Cardiovascular Health in an Australian Population"

_nutrients, 2023, doi:10.3390/nu15163645_

Round 1

Reviewer 1 Report

Specific Comments:

  • Figure 1 appears somewhat unintuitive and oversized given the information it contains. Consider incorporating a diagram, similar to Figure 1 from your previous publication (reference 28), to enhance clarity. Additionally, Figure 1's low quality, which may be attributed to the conversion into PDF format.

  • For Figure 4 (panels B-E), clarify whether the values on the x and y-axes are normalized in the figure description. Please provide units, if applicable.
  • While the discovery of correlations involving Butyricicoccus is noteworthy, it is essential to acknowledge that the correlation was not statistically significant after FDR adjustment (if I understood correctly). Additionally, as Butyricicoccus increases with age (10.1038/s41575-022-00605-x), potential age bias should be noted when interpreting these findings.

General comments:

The dataset appears comprehensive and well-defined, allowing for the possibility of conducting additional analyses. Consider exploring whether machine learning models, such as random forest, can differentiate between the Mediterranean and low-fat diets. Such an approach could unveil more intricate community signatures, not easily identifiable through the current statistical analysis.

Additionally, it might be informative to predict functional potential from the operational taxonomic units (OTUs) data. While acknowledging the need for cautious discussion, examining functional potential could provide unique insights, especially when dealing with microbiota from elderly individuals, which may already exhibit altered and potentially dysbiotic characteristics on taxonomic level.

The above suggestions are not mandatory for successful submission, but they could enrich the paper and offer further insights into the dataset.

Author Response

We thank the reviewer for the helpful comments and provide our point-by-point response to each comment below:

Figure 1 appears somewhat unintuitive and oversized given the information it contains. Consider incorporating a diagram, similar to Figure 1 from your previous publication (reference 28), to enhance clarity. Additionally, Figure 1's low quality, which may be attributed to the conversion into PDF format.
Response: We have amended Figure 1, including adjusting the scaling and improving the quality of the image.

For Figure 4 (panels B-E), clarify whether the values on the x and y-axes are normalized in the figure description. Please provide units, if applicable.

Response: Thank you for highlighting this. The values are based on the change (D) post intervention – pre intervention. We have now amended Figure 4B-E to clarify this and provided units for the measurement, where applicable.

While the discovery of correlations involving Butyricicoccus is noteworthy, it is essential to acknowledge that the correlation was not statistically significant after FDR adjustment (if I understood correctly). Additionally, as Butyricicoccus increases with age (10.1038/s41575-022-00605-x), potential age bias should be noted when interpreting these findings.

Response: Correlations between Butyricicoccus and Mediterranean diet adherence scores remained significant following FDR adjustment, but not with the clinical outcomes of systolic blood pressure and blood glucose. To provide clarity, we now provided both p-values and FDR p-values in Supplementary Table 2, in addition to highlighting this result in text.

As the reviewer has noted, it is possible that Butyricicoccus levels may increase with age, as shown previously. Our analysis indicating that an increase in Butyricicoccus relative abundance specifically by the MedDairy diet involved paired comparisons within subjects, which should obliterate the effects of age given the diet intervention of 8-weeks. In addition, the differences in Butyricicoccus levels were significant for between-group comparisons at the end of the diet intervention (post-MD vs post-LFD), despite no age difference between participants that were randomized to the MedDairy diet and the LFD diet. It may be possible that the extent of changes in Butyricicoccus abundance may be different in those of a different age demographic, such as a younger age cohort, given the potential for differences in baseline levels of Butyricicoccus compared to those observed in our study. We have revised our discussion to highlight this.

Line 485: “Previous studies have shown that levels of Butyricicoccus increased during ageing, and which replaced the loss of several commensal bacteria [64]. It is therefore possible that the impact of a MedDairy diet on Butyricicoccus, which was observed in this study, may differ in those of a different age cohort, such as a younger population.” 

General comments:

The dataset appears comprehensive and well-defined, allowing for the possibility of conducting additional analyses. Consider exploring whether machine learning models, such as random forest, can differentiate between the Mediterranean and low-fat diets. Such an approach could unveil more intricate community signatures, not easily identifiable through the current statistical analysis.

Response: We thank the reviewer for this suggestion. We agree that the abovementioned statistical tools may help in the exploration of specific variables that may contribute to group differences. However, developing machine learning models that incorporate complex interactions and non-linear relationships generally require either large datasets than the one available, or the use of additional data created through simulation. We will consider such statistical analysis in future studies that involve larger cohorts.

Additionally, it might be informative to predict functional potential from the operational taxonomic units (OTUs) data. While acknowledging the need for cautious discussion, examining functional potential could provide unique insights, especially when dealing with microbiota from elderly individuals, which may already exhibit altered and potentially dysbiotic characteristics on taxonomic level.

Response: We thank the reviewer for this suggestion. The prediction of functional potential, such as a PICRUSt analysis, infers functional traits based on the OTU information, which we are not able to substantiate further whether such inferences are valid. In our current study, we performed a comprehensive analysis to identify taxa that are differentially modulated by the MedDairy diet specifically. We then discussed the potential role of the taxa based on inference from studies that investigated the functional activity of these bacterial species. We therefore feel that functional prediction analysis does not provide further insight to the current analysis in our study.   

Reviewer 2 Report

The manuscript entitled „Interactions between Mediterranean diet supplemented with dairy foods and the gut microbiota influences cardiovascular health in an Australian population” presents interesting issue, but some problems should be corrected.

Major:

(1)    There is a problem with applied diets, which are described as ‘Mediterranean diet with 3-4 daily serves of dairy foods (MedDairy) or a low-fat (LFD) control diet’, but the indicated features (number of servings of dairy products and low fat content) are not opposite. One can imagine the decreased fat version of Mediterranean diet with adequate amount of dairy products. It is hard to understand that Authors compare low-fat diet (which should be characterized by adequate amount of dairy products – including low-fat dairy products) and diet with adequate amount of dairy. I suppose that Author should change the names of their diets to describe them properly, to make it understandable for readers.

(2)    It seems that Authors did not assess the dietary intake of their participants during the dietary recommendation, but they only assessed general adherence to Mediterranean diet (the necessary data are not presented). If they did not assess it, they can not conclude about the actual influence of the diet which was followed, as in fact they do not know the diet. It is the major limitation of the study. It Authors possess such data, they should calculate nutritional value of the diets.

General:

The manuscript is shabbily prepared, with e.g. unnecessary spaces, or missing spaces, or doubled numbering of references (the indicated issues are only some examples).

Abstract:

Authors should present the aim of their study (e.g. ‘The aim of the study was…’)

The characteristics of the studied group (inclusion criteria) should be presented in this section (an Abstract may be a little bit longer, if needed)

The brief characteristics of the applied diets is necessary.

Introduction:

Authors present a number of basic and even very trivial information that should not be presented in a scientific manuscript (e.g. “Diet is a fundamental determinant of metabolic health and immune regulation.”, or “The MedDiet delivers a range of bioactive nutrients including antioxidants, fibre, vitamins and minerals, polyphenols, monounsaturated fats, and omega-3 polyunsaturated fats”, or “Calcium is an essential mineral for the formation and maintenance of bone, but also for other biological processes”) – Authors should be aware that they do not prepare the basic manual for students, or column of the newspaper, but a scientific paper that should be interesting for researchers from the area of food and nutritional sciences, so they should understand that their readers will have the nutritional knowledge.

Authors formulate the opinion that Mediterranean diet contains 700 - 820 mg calcium per day with no reference presented. Authors should get familiar with various studies and present more balanced opinion.

Authors failed to justify the need for their study – they should present what is already known and what are the “gaps” in the scientific knowledge to formulate the aim of their study.

Authors should clearly justify compared diets.

Authors should present the aim of their study (e.g. ‘The aim of the study was…’)

Materials and methods:

Authors should clearly justify compared diets and present the clear description of the dietary recommendations, accompanied by nutritional value and servings of products for both diets.

Study assessment and outcomes – should be presented within Materials and methods sub-chapter

The nutritional value of the diet which was followed should be presented (see above)

It seems that Authors did not verify the normality of distribution for continuous variables – they should verify it, and indicate which statistical test was used for verification.

For normally distributed data Authors should present mean and SD values, but for the other distributions – present median, min and max values

Authors should use proper statistical test based on the observed distribution.

Results:

It seems that Authors did not verify the normality of distribution for continuous variables – they should verify it, and indicate which statistical test was used for verification.

For normally distributed data Authors should present mean and SD values, but for the other distributions – present median, min and max values

Authors should use proper statistical test based on the observed distribution.

Instead of figures, Authors should present tables, to be easier to follow.

Discussion:

Authors should not reproduce the information from the previous sections

The limitations of the study should be extensively discussed

Clear presentation of the general conclusion is needed.

Author Response

We thank the reviewer for the helpful comments and provide our point-by-point response to each comment in the document attached.
